# Alterations of Mitochondrial Network by Cigarette Smoking and E-Cigarette Vaping

**DOI:** 10.3390/cells11101688

**Published:** 2022-05-19

**Authors:** Manasa Kanithi, Sunil Junapudi, Syed Islamuddin Shah, Alavala Matta Reddy, Ghanim Ullah, Bojjibabu Chidipi

**Affiliations:** 1College of Osteopathic Medicine, Michigan State University, East Lansing, MI 48824, USA; kmk12345@gmail.com; 2Department of Pharmaceutical Chemistry, Geethanjali College of Pharmacy, Cherryal, Keesara, Medchalmalkajgiri District, Hyderabad 501301, India; suniljunapudi@gmail.com; 3Department of Physics, University of South Florida, Tampa, FL 33620, USA; islamuddinn@yahoo.com; 4Department of Zoology, School of Life and Health Sciences, Adikavi Nannaya University, Rajahmundry 533296, India; alavalareddy@hotmail.com; 5Morsani College of Medicine, University of South Florida, Tampa, FL 33612, USA

**Keywords:** cigarette smoking, e-cigarette smoking, mitochondria, fusion, fission

## Abstract

Toxins present in cigarette and e-cigarette smoke constitute a significant cause of illnesses and are known to have fatal health impacts. Specific mechanisms by which toxins present in smoke impair cell repair are still being researched and are of prime interest for developing more effective treatments. Current literature suggests toxins present in cigarette smoke and aerosolized e-vapor trigger abnormal intercellular responses, damage mitochondrial function, and consequently disrupt the homeostasis of the organelle’s biochemical processes by increasing reactive oxidative species. Increased oxidative stress sets off a cascade of molecular events, disrupting optimal mitochondrial morphology and homeostasis. Furthermore, smoking-induced oxidative stress may also amalgamate with other health factors to contribute to various pathophysiological processes. An increasing number of studies show that toxins may affect mitochondria even through exposure to secondhand or thirdhand smoke. This review assesses the impact of toxins present in tobacco smoke and e-vapor on mitochondrial health, networking, and critical structural processes, including mitochondria fission, fusion, hyper-fusion, fragmentation, and mitophagy. The efforts are focused on discussing current evidence linking toxins present in first, second, and thirdhand smoke to mitochondrial dysfunction.

## 1. Introduction

Cigarette smoking (CS), e-cigarette (EC) vaping, and other types of exposure to environmental tobacco smoke, including second and thirdhand smoke, are dangerous to human health, causing diseases that affect every organ system [1,2,3,4]. Despite thorough documentation of the extensive damages caused by smoking, it continues to be one of the most prevalent public health concerns worldwide, claiming millions of lives each year [2]. Primary and secondary exposure to tobacco smoke significantly raises the risk of cancer [5,6,7], coronary heart disease [8,9,10] stroke [11,12], and bacterial and viral infections, and has detrimental impacts during pregnancy [13,14,15].

The detrimental effects of tobacco smoke are not limited to smokers. Non-smokers exposed to second and thirdhand smoke in the environment also show an increased risk for health concerns [4]. Cigarette smoke is composed of thousands of chemicals, of which many are volatile, carcinogenic, and cause DNA damage [16,17,18]. These chemicals can reside in the environment until inhaled by non-smokers to continue causing devastating health impacts. Secondhand smoke (SHS) is a mixture of what is exhaled by the smoker and what is emitted by the burning tobacco product, whereas thirdhand smoke (THS) is the residue that accumulates from SHS deposited onto surfaces and can exist in the environment for a long time.

E-cigarettes, also known as electronic nicotine delivery systems (ENDS), were first introduced and marketed as a healthier alternative to cigarettes and to help with smoking cessation [19]. However, new dangers continue to present themselves as ENDS use continues to rise among young adults and adolescents [20]. E-cigarette liquid (e-liquid) is composed of various agents, typically a solvent, nicotine and additional flavoring compounds suspended in a humectant inhaled as an aerosol [21]. Despite the original intention to reduce the detrimental impacts of tobacco smoking, ENDS are largely unregulated and pose increasing concerns as they are proving to be toxic for neonates [19,22], vascular health [23,24], respiratory health [25], and damaging to the oral cavity [26].

This paper discusses the pathophysiological mechanisms by which toxicants from tobacco smoke and ENDS negatively impact mitochondrial function. Mitochondria play a vital role in redox signaling [27], cell cycle regulation, differentiation, DNA exchange between cells to restore function, and apoptosis. As a result of their critical role in cell survival and proliferation, mitochondria are often an organelle of interest when studying cellular mechanisms impacting multiple disease processes. Furthermore, mitochondrial susceptibility to damage by exposure to toxicants justifies investigating the impact of environmental chemicals on the organelle [28]. The effects of inhaled toxicants can include inhibition of ATP synthesis due to uncoupling of oxidative phosphorylation [29], increased oxidative mitochondrial damage [30], and mitochondria-initiated apoptosis [31].

Disruptions in mitochondrial function can result from impaired morphology, fusion and fission imbalances, increased oxidative stress, or even mutations within the mitochondrial DNA (mtDNA). Impaired mitochondrial morphology and function are implicated in the pathogenesis of pulmonary [32], neurodegenerative [33], and diabetic kidney diseases [34], and can be pro-tumorigenic [35], and potentiate inflammatory responses [36]. Mitochondrial DNA, a circular chromosome found within the organelle maintained by nuclear-encoded proteins [37], is another component that can contribute to disease if damaged [38]. Mutations in mitochondrial DNA are responsible for numerous diseases, such as optic atrophy [39], mitochondrial myopathy [40], and diabetes mellitus [41].

## 2. The Mitochondrial Fusion and Fission Machinery

The mitochondria’s dynamic and healthy nature is dependent on inter-organelle crosstalk and key processes, including mitochondrial fusion, fission, mitoptosis, and mitophagy [42,43,44,45]. The joint forces of mitochondrial fusion and fission, maintained by dynamins and mitofusins, support robust and optimal network morphology [29,46,47], and respond to physiological conditions [48]. On the contrary, an imbalance in the proteins determining mitochondrial dynamics can aggravate type 2 diabetes [49], neurodegenerative diseases [50,51], and can be embryonically lethal [52,53].

Defective mitochondria are disposed of through mitochondrial fission [54], whereas mtDNA exchange and rescue of damaged mitochondria can happen through fusion [55]. The dynamin superfamily contains a variety of ubiquitous GTPases involving multiple processes, including regulating mitochondrial fission. Dynamin-related protein 1 (Drp1) is known to play a critical role in mitochondrial homeostasis by forming fission rings to eliminate damaged parts of the mitochondrial membrane. Drp1 binding requires the presence of different proteins, including Mitochondrial Fission 1 Protein (FIS1) on the outer mitochondrial membrane, mitochondrial dynamics proteins 49kDa (MiD49), 51 kDa (MiD51) [56], and mitochondrial fission factor (MFF) [57]. Mitochondrial fusion is dependent on the presence of another class of GTPases known as mitofusins [58]. Mitofusins 1 and 2, located at the outer mitochondrial membrane, reciprocally interact with Drp1 to enable fusion [59], support proper embryological development and enhance mitochondrial cooperation to prevent respiratory dysfunction [53]. The mitochondrial dynamin-like GTPase, Optic atrophy 1 (OPA1), is located on the inner mitochondrial membrane (IMM). OPA1 exists in many isoforms that work together to induce mitochondrial fusion [60]. When there is a loss of membrane potential longform OPA1 isoforms become destabilized and can no longer optimally function [60]. We demonstrate mitochondrial fusion and fission in cartoon Figure 1.

Apart from fusion and fission, mitophagy is another critical process that degrades mitochondria through autophagy. PTEN-induced putative kinase 1 (Pink1) initiates the clearance of defective mitochondria through mitophagy. This clearance pathway is initiated in various pathological conditions [61,62,63] to protect the organelle from dysfunction. Pink1 accumulates on the outer mitochondrial matrix of dysfunctional mitochondria. The increased Pink1 concentration consequentially recruits Parkin, an E3 ubiquitin ligase. The Pink1/Parkin complex acts as marker signaling the defective organelle for clearance by mitophagy. Consequentially, mitophagy is quintessential to protecting optimal mitochondrial form and function and operates on multi-tiered processes [64,65]. Failure to initiate mitophagy, or reduction in its process, promotes mitochondrial oxidative stress. This failure sets off a cascade of imbalanced signaling pathways, which can accelerate disease processes, including neurodegenerative and cardiovascular conditions [66,67,68].

## 3. Cigarette Smoke and E-Cigarette Vape Extraction

In vitro effects of cigarette smoke and e-cigarette vapor are best studied through cell culture media, and consequently various systems have been developed over time to accommodate studies conducted in larger or smaller chambers. We will quickly review two examples of how extraction systems are set up. SV Teague et al. developed a method for smaller chambers that is specifically useful for maintaining consistent levels of total suspended particles to replicate better relevant environmental conditions of smoke exposure [69].

CS or EC vaping machines contain cigarette or EC holding and lighting devices. A metered puff controller regulates the number of smoke or vape puffs through a flow machine, and chimneys in the conditioning chamber dilute smoke puffs to distribute smoke to each chamber containing animal or cell-culture media [70]. Abouassali O et al. extracted e-vapor in cell culture media to study the in vitro toxicity of flavored e-liquids [70]. A 10-cm × 10-cm × 7-cm chamber was made with a bottom opening fitted for the vaping device’s mouthpiece. The chamber contained inlet and outlet openings on the top of the lid. The inlet tube received air through an air pump connected to a flow meter, whereas the outlet tube delivered the e-vapor to the cell culture media. Regulating the vacuum connected to the flow meter enabled control of puff size and led to vapor bubbling into the cell culture medium [70].

### 3.1. Cigarette Smoke (CS) and Cigarette Smoke Extract (CSE) Trigger Mitochondrial ROS

Mitochondrial reactive oxygen species (mtROSs) are invaluable intermediates of cell signaling pathways. Their production integrates various biochemical processes to sustain cell survival, signaling, and energetics [71]. In addition to making ATP, the electron transport chain maintains the organelle’s electrochemical potential, which signals the proper function and integration of other metabolic and cellular processes. Disruption in one or more of the processes supporting healthy mitochondrial populations and networking has multifaceted implications and can disrupt the organelle’s ability to regulate ROS levels. In excess, mtROS can contribute to mitochondrial dysfunction and many diseases [72,73,74]. This undeniable influence of mtROS on other signaling pathways is a key mechanism often implicated by inhaled toxicants [32,75,76,77].

Wang Z et al. found that CS increased oxidative stress, reduced respiration, and disrupted the balance of mitochondrial fusion and fission, resulting in altered mitochondrial morphology in primary rat lung microvascular endothelial cells (LMVEC) [78]. One of the ways CS altered mitochondria was by shortening the organelles’ network and making it smaller. These smaller networks of damaged mitochondria were more prone to perinuclear accumulation. CS disrupted the balance of fission and fusion events by upregulating mitochondrial fission and decreasing fusion. Increased mitochondrial fission resulted from decreasing Drp1-S637 and increasing FIS1, Drp1-S616 phosphorylation [78]. On the other hand, CS-mediated Mfn2 reduction in LMVEC and mouse lungs resulted in reduced mitochondrial fusion. In addition to altered fission and fusion events, CS also induced mitochondrial translocation and tetramerization [78].

In healthy non-smokers, acute THS exposure increased oxidative stress and upregulated ROS scavenging genes in human nasal epithelial cells [79]. Compared to healthy non-smokers, smokers had increased somatic mtDNA mutations in their buccal cells [80]. Not only is mtDNA susceptible to oxidative damage, but the mutations themselves can target the cytochrome c oxidase subunit 1 complex [80]. Cigarette smoke extract (CSE), in vivo, has been shown to increase oxidative stress and reduce mitochondrial membrane potential in human fetal fibroblast strains and human lung fibroblast HFL-1 and L828 [81]. Morphologic characteristics of apoptotic cell death were seen after incubating fibroblasts with 10% CSE for six hours [81]. Exposure to 5% CSE at 3 h initially reduced glutathione (GSH) levels, consequently reducing the organelle’s antioxidant capacity. The lipophilic components of CSE disrupt mitochondrial function in bronchial epithelial cells by causing a dose-dependent decrease in mitochondrial membrane potential and intracellular ATP levels while also increasing ROS [82]. Low dose CSE triggers proadaptive survival mitochondrial hyper-fusion in mouse alveolar epithelial cells. These changes appear to play a protective role in maintaining the organelle’s function during initial stages of exposure [83]. These structural changes were seen with increased levels of MFN2 within 24 h of 10% or 20% CSE treatment [83].

Interestingly enough, these proadaptive mitochondrial morphology changes were accompanied by increased metabolic activity and ATP levels without causing an increase in mitochondrial superoxidases [83]. The studies thus far indicate the important role that dosage and exposure to CS play. Initial responses triggered by CSE are primarily protective. However, these protective changes can lead to irreversible damage or exacerbate pathological processes in dose and time-dependent manners.

CSE exposure in human bronchial epithelial cells (HBECs) caused mitochondrial fragmentation and increased mitochondrial ROS production, leading to an increased percentage of cellular death [84]. Hara et al. found that the HBEC mitochondria in COPD lung tissue were more prone to fragmentation, indicating that preexisting conditions increase the organelle’s susceptibility to damage by CSE. Whole cigarette smoke condensates (WCSCs) decreased cell viability in the normal human bronchial epithelial cell line (BEAS-2B) in a dose-dependent manner and disrupted mitochondrial homeostasis by inducing hypoxic conditions [85]. WCSC caused a clear dose-dependent elevation of IL-6 and IL-8, and increased ferritin protein expression. Increased levels of apoptosis, autophagy, ER stress, antioxidant, and MAP kinase activation-related proteins suggested WCSC may induce ferroptosis through disrupting homeostasis.

In human airway smooth muscle, CSE increased Drp1 and reduced mfn2 in a concentration-dependent fashion [59]. These changes induced mitochondrial fragmentation and damaged morphology by reducing mitochondrial branching and branch length [59]. Furthermore, Aravamudan et al. showed how fusion-fission protein disruption could negatively influence ROS production, cell proliferation, and apoptosis in airway diseases, such as asthma and chronic obstructive pulmonary disorder (COPD) [59]. Long-term CSE in COPD primary bronchial epithelial cells induced mitochondrial fragmentation and altered morphology by reducing the number of cristae [86]. These changes in morphology were noted alongside increases in oxidative stress markers, OXPHOS proteins, proinflammatory mediators, and expression of fission and fusion markers.

### 3.2. The Role of Nicotine in Mitochondrial Dysfunction

Nicotine is richly present in tobacco products. Nicotine is one of the many thousands of compounds present in cigarette smoke and is commonly found in ENDS [87]. CS and ENDS produce nicotine boli which are transported to the central nervous system and activate nicotinic acetylcholine receptors (nAChRs) [87], present throughout the body. The interaction between nicotine boli and nAChRs can modulate mitochondrial dynamics in hippocampal neurons [88], in lung cancer [89,90], and can impact fetal and neonatal development [91].

A previously done western blot analysis revealed that 10 µM of nicotine induced mitochondrial fragmentation in human multipotent embryonal carcinoma cell line NT2/D1, by significantly decreasing Mfn1 and Mfn2. This effect is based on an unknown mechanism [92]. Hirata et al. confirmed the mechanism by using a nonselective nAChR antagonist, which effectively blocked nicotine-induced reduction of Mfn1 and Mfn2 protein levels, ATP levels, and mitochondrial fragmentation. Guo et al. proved that in non-small cell lung cancer cells, nicotine-induced activation of hypoxia-inducible factor (HIF)-1α was dependent on mitochondrial-dependent ROS activating downstream Akt and MAPK signaling pathways, as well as transcriptional regulation, via factors such as NF-κB and nuclear erythroid 2-related factor 2 [59,89]. The CSE decrease of mitochondrial fusion by decreasing Mfn2 and elevating fission by elevating Drp1 causes mitochondrial fragmentation. Optimal mitochondria numbers are required to generate the ATP for cellular demand, and an imbalance of mitochondrial fusion and fission elevates cellular ROS. We demonstrate this in Figure 2. Maternal nicotine, regardless of cigarette smoking or nicotine replacement therapy, induces oxidative stress targeting the mitochondria, as well as β-cell apoptosis in the pancreas, negatively impacting the offspring [91].

### 3.3. Constituents of Fluids Used in ENDS Are Cytotoxic and Impair Mitochondrial Function

As the number of flavors available for e-cigarettes continues to increase, more studies must be done to understand the impact of various flavors and solvents on health [94]. Currently, studies present evidence for the varying toxicity of different flavors, indicating the need to generate a profile of which compounds could be more toxic [70,95,96,97]. Specific studies exploring the toxicity of flavors on mitochondria are limited. However, some of these flavors and their toxicities will be discussed in this section before further expanding on studies specific to mitochondrial function.

Cinnamaldehyde or vanillin-flavored e-vapor was toxic in HL-1 cardiomyocytes and compromised cardiac electrophysiology [70]. Abouassali et al. set up a vaping chamber that expelled a puff volume of 110 mL set on a cyclical timer to mimic ENDS user consumption. After ten weeks of inhaling vanillin-aldehyde e-vapor, an increased sympathetic variance was noted in heart rate. In vivo inducible ventricular tachycardia and the magnitude of ventricular action potential duration alternans were respectively more prolonged and larger in vanillin-flavored e-vapor exposed mice. HL-1 atrial cardiomyocytes are more susceptible to necrosis and apoptosis when cultured with vanilla-custard e-vapor extract in a concentration-dependent manner. Mitochondria play a critical role in maintaining cardiomyocyte homeostasis. Therefore, more studies exploring the mechanistic connection between the impact of e-vapor on mitochondria and cardiomyocytes could prove invaluable. H292 human bronchial epithelial cells exposed to strawberry-flavored e-vapor had reduced metabolic activity, reduced cell viability, and increased interleukin release [95]. Cooper et al. demonstrated how vaping-related reinforcement behavior is elevated in male mice when self-administering menthol or green-apple flavored e-liquid compared to no flavor e-liquids [96]. Farnesol, a component of green apple flavor, has been shown to significantly increase nicotine-reward related behavior by altering baseline firing of GABA neurons and upregulating nAChR function, especially in male mice [97]. Interestingly enough, although ENDS are commonly advertised as a safer alternative to cigarette smoking, ECE was found to cause cardiomyocyte toxicities and generate oxidative stress similar to CSE [98]. Jabba et al. demonstrated that solvent adducts of reactive flavor aldehydes are more cytotoxic on lung epithelial cells (BEAS-2B and A549) than their parent aldehydes, due to the rapid chemical reaction they undergo with e-liquid solvents, propylene glycol, and vegetable glycerol (PG/VG). Furthermore, these reactive flavor aldehydes reduced mitochondrial ATP synthesis and were more cytotoxic than their parent aldehydes [99]. When compared to parent aldehydes, benzaldehyde PG, vanillin PG, and ethyl vanillin PG reduced mitochondrial oxygen consumption rate in a concentration-dependent manner with more potency [99]. This suggests the importance of looking at secondary reaction products and not just the parent compound when assessing cytotoxicity of EC fluids. Williams et al., demonstrated that MTT assay marked the cytotoxicity of two potent chemical toxins in EC solvents, selenium and arsenic. Both chemicals inhibited mitochondrial reductases in BEAS-2B cells and proved toxic for pulmonary fibroblasts, whereas selenium increased superoxide production in mitochondria [100]. In this particular study, selenium was present in all products, whereas arsenic was present in a few. This consideration brings awareness to a major concern regarding the variability in metals and metalloid concentrations present in EC liquid. This variability depends on the manufacturing company, packaging, coil type, and where the EC was purchased or obtained [101]. Depending on the user, the amount of vapor inhaled can also vary. This lack of consistency points to a significant need for regulation around manufacturing and distribution protocols to better understand the risk of exposure to different metals.

## 4. Maternal Health

The dangers of smoking during pregnancy have long been elucidated, and smoking cessation is advised for the duration of pregnancy. If smoking cessation is impossible, replacing cigarettes with EC is generally advised. There is a prevalent perception of ENDS being safer than smoking cigarettes. However, their use must be monitored during pregnancy [102]. Currently, there is insufficient data to understand the impact of vaping on offspring due to variability in ENDS usage amongst pregnant women [103]. Li et al. exposed mice to either air or cigarette smoke from six weeks before pregnancy until lactation. At mating, a subset of the mice exposed to CS were then shifted over to e-vapor exposure. The mice exposed to cigarette smoke for the entire duration had offspring with impaired glucose tolerance, increased plasma non-esterified fatty acids and liver triglyceride concentrations [104]. The offspring born to mice that had EC exposure upon mating did not improve glucose tolerance; however, these offspring did have reduced toxicity in pregnancy and reduced hepatic lipid metabolism [104]. Female Balb/c mice exposed to e-vapor, with and without nicotine, for six weeks before mating showed detrimental changes along with their offspring [104]. The offspring exposed to nicotine-free vapor had metabolic changes and liver damage, while those exposed to nicotine vapor had liver steatosis [104]. Nicotine and e-cigarette condensate have been shown to disrupt mitochondria by inhibiting OXPHOS complex III and increasing mtROS [105]. This increased mitochondrial dysregulation in redox signaling was ensued by inhibition of myofibroblast differentiation critical for proper development in HLF-1, which further impaired wound healing [105]. A previous assessment of in vitro and animal models highlighted the multifaceted mechanisms through which ENDS impacted pre and postnatal brain development [106]. Zahedi et al. demonstrated that the mechanism behind only EC with low or zero nicotine levels in EC -induced stem cell toxicity is stress-induced mitochondrial hyperfusion (SIMH), resulting in transitory survival, followed by increasing mitochondrial oxidative stress [107]. SIMH is identified as a survival response to nicotine and is largely present in EC refill fluids called do-it-yourself EC products [107]. They also observed that a high nicotine concentration (110 μg/mL) with EC caused a rapid influx of calcium. EC leads to cellular stress, diminishes cellular health in the stem cell population, elevates cellular aging, and develops mitochondriopathies [107]. We illustrated the EC effect on mitochondrial function in Figure 3. The susceptibility of stem cells, specifically neural stem cells, is essential to consider during embryonic development. Compared to adult lung cells, embryonic and neonatal cells were more sensitive to EC refill fluids [108]. This sensitivity was correlated to the number and concentration of chemicals used to flavor the refill fluids [108]. Since ENDS are marketed as a “safer” alternative to cigarettes, the connection between chemicals used for flavoring and cytotoxicity is worth paying attention to as these refill fluids are not currently regulated and have not been fully assessed for their effects. Behar et al. narrowed down cinnamaldehyde (CAD) and 2-methoxycinnamaldehyde (2MOCA) as highly cytotoxic to human embryonic stem cells [109]. These were two of the most toxic ingredients found within cinnamon-flavored EC refill fluids. The identification of toxicants in different flavors across different manufacturing companies is lacking. However, to fully understand the impact ENDS could have on fetal development and offspring health, this is an area of research that must be expanded. Maternal vaping during pregnancy may not be as extensively studied as maternal smoking during pregnancy but is a critical and imperative area to be considered when designing further studies.

## 5. Thirdhand Smoke

While the health impacts of thirdhand cigarette smoke (THS) are still being studied, it is undeniably a crucial component of addressing public and environmental health. Although experiments on the hazardous nature of THS are limited, an emerging set of data provide reasons not to exclude THS from future studies assessing risk factors. THS has been shown to cause stress-induced mitochondrial hyperfusion (SIMH) in mouse neural stem cells along with increased mitochondrial membrane potential (MMP), increased ATP levels, increased superoxide production, and increased oxidation of mitochondrial proteins [110]. SIMH can also dysregulate gene regulation, and transcription has been shown to reduce mitochondrial fission protein Fis1 expression [110], Figure 4 In Vivo exposure to THS increased AST, urea, and nuclear respiratory factor-1 (NRF1) levels in a time-dependent manner leading to increased liver dysfunction and mitochondrial dysfunction within the liver [111]. THS exposure has been shown to reduce the liver’s antioxidant potential in a time-dependent manner. This reduction in liver function was seen with some key changes, such as increased oxidative stress, reduced ATP levels, and increased lactate, indicating mitochondrial dysfunction within the liver [111]. Adhami et al. proposed that the connection between the dysfunctional changes was due to significantly higher TNF-α levels, which could play a role in mitochondrial dysfunction, given the cytokine’s role in many inflammatory and cell death pathways [111].

Pozuelos et al. conducted a randomized control trial to demonstrate how the acute inhalation of THS increased oxidative stress, mitochondrial membrane potential, ATP production, and decreased permeability of transition of mitochondria and mitochondrial membranes. These changes were accompanied by stress-induced mitochondrial hyperfusion and dysfunction and increased DNA repair mechanisms [79,112]. Stem cells exposed to THS also showed an increase in SIMH, but upon prolonged exposure, both mitochondrial membrane potential and cell proliferation decreased, ultimately leading to apoptosis of the cell [110].

To summarize, cigarette smoke and e-cigarette vapor are culprits causing many devastating and fatal diseases. Although tobacco smoking increases the risk of contracting diseases, psychological and social factors play a key role in maintaining the habit [113]. With an ever-present concern for this public health crisis, toxins in tobacco smoke have an undeniably detrimental effect on mitochondrial health further aggravating the pathophysiological mechanisms of different diseases. Understanding these mechanisms can be helpful in the development of therapeutics. For example, iPSC-MSCs reduced airway inflammation and offered protection against CS-induced mitochondrial oxidative stress, dysfunction, and apoptosis in human ASMCs and mouse lungs [114].

## 6. Conclusions

This review highlights how mitochondrial damage caused by inhaled intoxicants increases ROS production, apoptosis, reduces respiration, alters mitochondrial membrane potential, and destroys the equilibrium of fission/fusion effects. These detrimental changes contribute to aggravated inflammatory pathways and various disease pathogeneses. Mitochondrial damage responses to smoke vary in tissue-dependent and concentration-dependent manners, which indicate a need to develop specific studies on understanding the diverse effects and mechanisms which contribute to each unfolding disease process. Understanding these mechanisms can aid in the development of effective interventional and therapeutic modalities.

Furthermore, mitochondrial morphology and health are maintained by the dynamic opposing forces of mitochondrial fusion and fission, which are altered by means of various mechanisms involving increased mtROS, increased ATP, dysregulation of key proteins, and stress-induced mitochondrial hyperfusion. The intimate connection between mitochondrial morphological changes and dysfunction impairs multiple pathways and alters downstream signaling. Interestingly, these toxic changes vary based on the chemical composition of different e-liquids. The cell type also may impact which changes occur first and how mitochondria can behave differently in acute versus long-term exposure. Another factor influencing organelle behavior is exposure type and whether or not the exposure was direct or through second or thirdhand smoke. Therefore, further research on the cytotoxicity of aldehyde flavors available for ENDS users is crucial to developing public health guidelines to elucidate specific pathophysiological mechanisms. Although ENDS devices are offered as healthier alternatives to cigarette smoking, it is clear that they are not without risks, especially when concerning maternal health.

## Figures and Tables

**Figure 1 cells-11-01688-f001:**
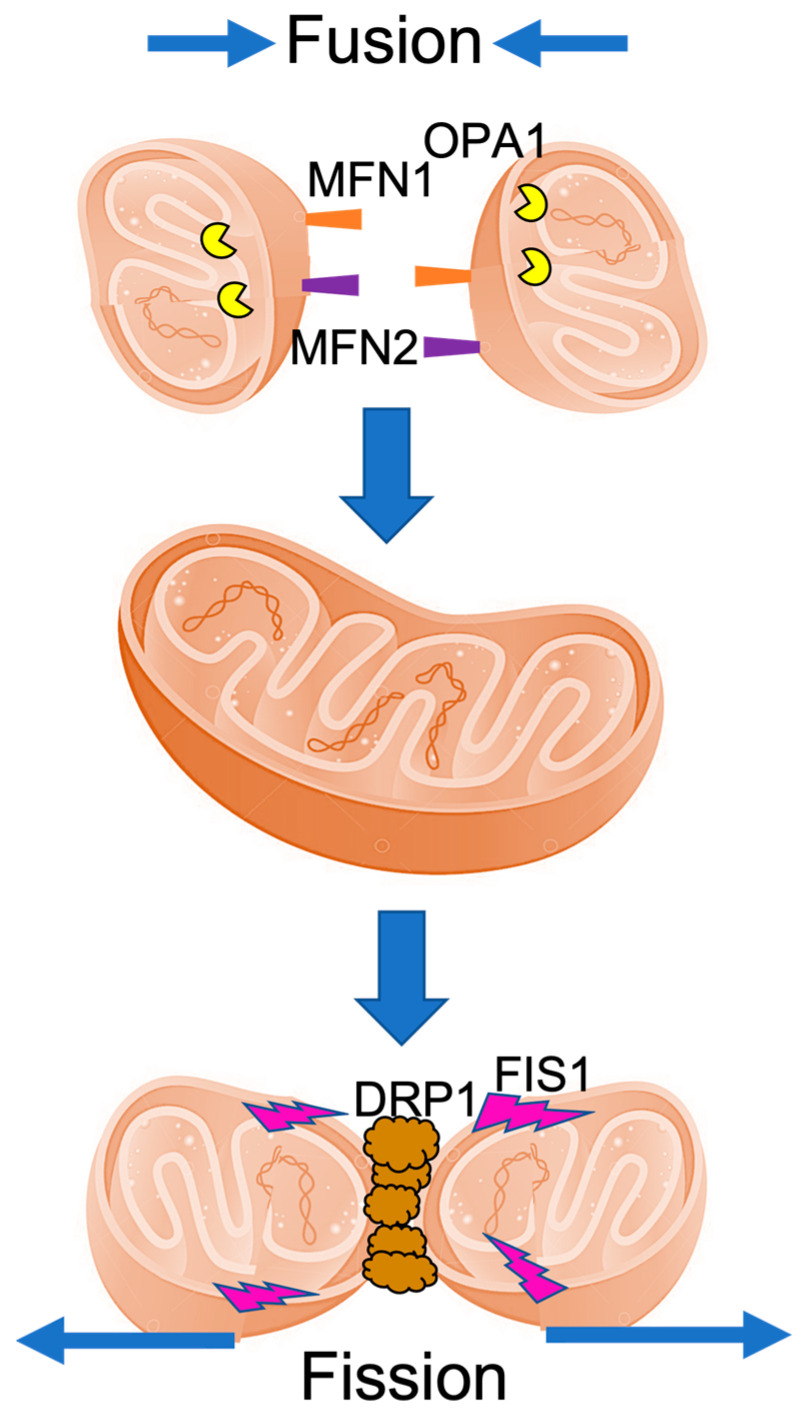
Schematic cartoon of mitochondrial fusion and fission. Mitochondrial fusion occurs when two mitochondria fuse together, whereas fission occurs when one mitochondrion splits into two. Fusion is coordinated on the OMM by the mitofusins (MFN1 and MFN2), and on the IMM by optic atrophy 1 (OPA1). In fission, FIS1 and drp1 are largely involved.

**Figure 2 cells-11-01688-f002:**
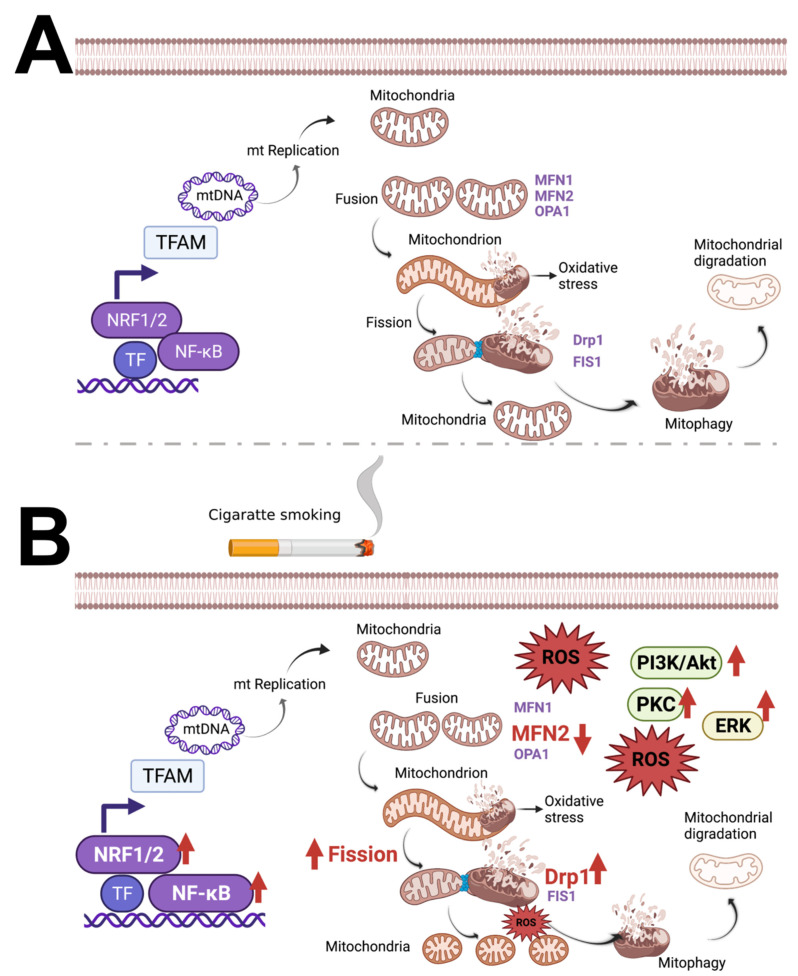
Mitochondrial quality control pathways in healthy cells and CSE treated cells. (**A**). Mitochondrial biogenesis and network regulation rely on key transcription factors and various proteins to maintain the processes needed for the organelle’s homeostasis. Mitochondrial transcription factor A (TFAM) acts on mtDNA after being imported into mitochondria and is an essential transcription factor needed to encode mitochondrial proteins. The first of these proteins transcribed include electron transport chain subunits to increase and maintain the appropriate number of mitochondria to sustain oxygen consumption and ATP synthesis. Secondly, fusion proteins (MFN, MFN2, and OPA1) and fission proteins (Drp1 and FIS1) are transcribed to maintain healthy morphology by excising and clearing out damaged portions of the organelle. Mitophagy, a specialized autophagic pathway, eventually recycles the discarded mitochondrial components [93]. (**B**). In human airway smooth muscle, CS disrupts mitochondrial homeostasis by causing morphological changes and dysfunction. CS-induced mitochondrial fragmentation and damage to networked morphology occurs in a concentration-dependent fashion. CS also increased Drp1 expression, decreased Mfn2, and involved ROS. Furthermore, NF-κB and nuclear erythroid 2-related factor 2 (NRF2) lead to a transcriptional upregulation and increased activation of extracellular signal-regulated kinase (ERK), phosphatidylinositol 3-kinase (PI3K)/protein kinase B (Akt), and protein kinase C (PKC) [59].

**Figure 3 cells-11-01688-f003:**
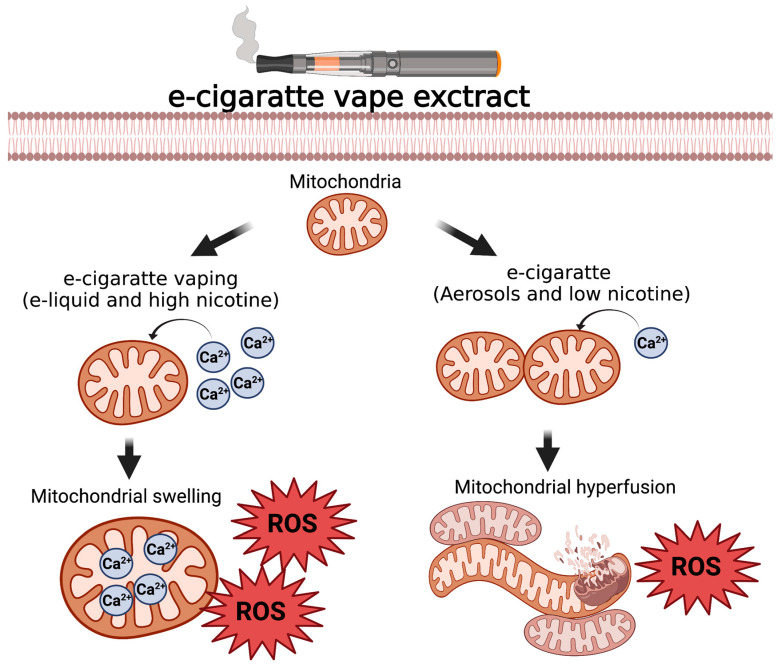
The effects of EC vaping on mitochondrial function. EC vaping with high concertation of nicotine causes mitochondrial swelling associated with mitochondrial calcium overload and increases ROS. EC vaping with aerosols and low nicotine concentration causes mitochondrial hyperfusion associated with stress and elevates the ROS.

**Figure 4 cells-11-01688-f004:**
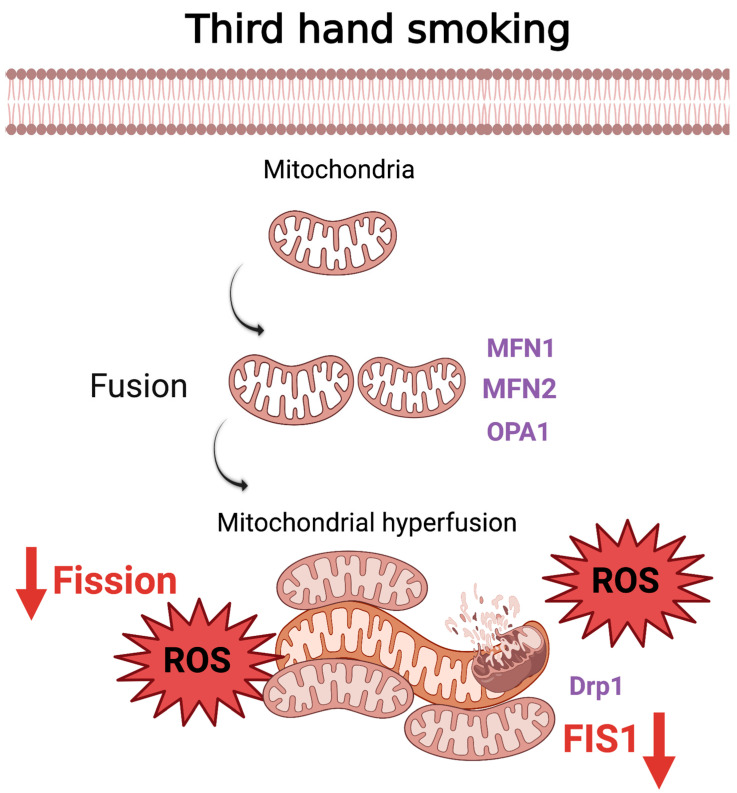
The effect of THS on mitochondria and cell health. THS caused SIMH accompanied by decreased expression of mitochondrial fission protein Fis1 causes the elevation of ROS [110].

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
