# Peer review of "Alterations of Mitochondrial Network by Cigarette Smoking and E-Cigarette Vaping"

_cells, 2022, doi:10.3390/cells11101688_

Round 1

Reviewer 1 Report

Kanithi and colleagues summarize recent evidence linking cigarette and e-cigarette use to altered mitochondrial function.  The topic is of general interest, and the manuscript is overall well written, although I outline a few suggestions to improve organization and flow.

Comments:

Could you please explain your rationale for including a subsection and figure on a specific in vitro model?  It is not a new method, and it is disconnected from the rest of the review.  Please either better integrate or remove.

There are two issues that arise throughout the manuscript—please find instances throughout and revise:

-Descriptions of specific studies are often confusing to follow throughout the manuscript.  One good example is lines 159-162.

-Many paragraphs are structured around individual studies (for example line 177) or a series of observations (for example line 216) with no further interpretation.  Readers would benefit from some small changes, for example incorporating topic and summary sentences into paragraphs, as well as some additional discussion.

What is being depicted for the mitophagy-targeted mitochondrion in figure 3? Bursting?

The last two paragraphs of the introduction contain multiple redundant sentences.  Please revise.

Please revise line 149: “The four parts of the electron transport chain make ATP”

Please revise or better support line 117-119.

Line 99: The role of mitofusins in mitochondrial fission is unclear as written and not supported by references 59-61.

Author Response

It is with excitement that we resubmit to you a revised version of manuscript cells-1708425, the manuscript entitled “Alterations of Mitochondrial Network by Cigarette smoking and e-Cigarette Vaping” for the Cells journal. Thank you for giving me the opportunity to revise and resubmit this manuscript. We appreciate the time and effort that the editor and the reviewers dedicated to providing suggestions for improvement. We have revised the manuscript, and as a result of the reviewers’ recommendations which we addressed, the manuscript has been dramatically improved. We highlight in yellow the changes that were made to the manuscript. Please see below for a point-by-point response to the reviewers’ comments and concerns.

Reviewer 1:

Kanithi and colleagues summarize recent evidence linking cigarette and e-cigarette use to altered mitochondrial function.  The topic is of general interest, and the manuscript is overall well written, although I outline a few suggestions to improve organization and flow.

Comments:

Could you please explain your rationale for including a subsection and figure on a specific in vitro model?  It is not a new method, and it is disconnected from the rest of the review.  Please either better integrate or remove.

 Response: Now we removed Filgure.2

There are two issues that arise throughout the manuscript—please find instances throughout and revise:

-Descriptions of specific studies are often confusing to follow throughout the manuscript.  One good example is lines 159-162.

Response: Now we reconstructed the text. Thank you

-Many paragraphs are structured around individual studies (for example line 177) or a series of observations (for example line 216) with no further interpretation.  Readers would benefit from some small changes, for example incorporating topic and summary sentences into paragraphs, as well as some additional discussion.

Response: The studies discussed in Section 3.1, which contained the lines mentioned in the comments above, were expanded upon. Added additional content to explain the studies further and why these findings may be of importance. Thank you.

What is being depicted for the mitophagy-targeted mitochondrion in figure 3? Bursting?

Response: Mitophagy is the end stage of mitochondrial degradation. In figure 3, we focused on aberrant mitochondrial fusion and fission proteins resulting in mitochondrial fragmentation. Please refer to lines from

The last two paragraphs of the introduction contain multiple redundant sentences.  Please revise.Please revise line 149: “The four parts of the electron transport chain make ATP”Please revise or better support line 117-119. Line 99: The role of mitofusins in mitochondrial fission is unclear as written and not supported by references 59-61.

Response: Revisions were made for all four comments above accordingly. Excessive sentences in the introduction were removed. Lines 149 and 117-119 were revised with better explanations in order to clarify them.  and Line 99 regarding the role of mitofusin were corrected so that it is now supported by references

Reviewer 2 Report

Kanithi et al. review the evidence that CS and ECV alter mitochondrial dynamics and ROS output. While the evidence in the case of CS is clear, less is known about ECV. Mitochondria are a main control unit of cellular redox and oxidative stress, making them particularly vulnerable to toxic agents found in either way of nicotine consumption and possibly nicotine itself. Mitochondrial dynamics are one possible readout of mitochondrial damage.

The fission and fusion of mitochondria is a clearly defined cell biological mechanism that has mechanistic triggers and inhibitors. Other than poorly described phenomenological observations, the review currently does not provide much insight into how these are altered and how likely that these pathways are altered in users. The authors present a novel topic that is of interest to the public and scientists but their text barely scratches the surface of the cell biology that it aims to review. Generally speaking, the review should distinguish the molecular effects of nicotine versus the effects of additives. And it must describe the mechanisms behind it, including concentrations used for individual observations. This could put them into context into what happens in users. In its current form, the review is of little use to readers who would like to know more about the mechanistic background. 

Specific points:

  1. Regarding any effects of nicotine on mitochondria, it should be mentioned whether the compound was added at concentrations resembling what is found in consumers.
  2. The cell biological mode of action of nicotine is not described. How does it interfere with mitochondrial pathways? How does it trigger Mfn1/2 degradation? It is mentioned that it acts on nAChRs, but this is unlikely causing effects in mitochondria. This is the core of the review but the text provides little to no information.
  3. The toxicity of certain flavors in ECV is mentioned. Are these present in sufficient amounts in users? The review proposes strict regulation but does not give much insight into what was tested and why mitochondria are impacted.
  4. The presence of selenium and arsenic is mentioned in vaping solvents. Are these always present or are they an occasional contamination?
  5. What detrimental changes in offspring are observed with ENDS? Why? Are they restricted to ENDS containing copper microparticles? How often are these used by manufacturers?
  6. Some tests result in mitochondrial hyperfusion, others in mitochondrial fragmentation. These are fundamentally different behaviors, suggesting different experimental inputs have been used but they are not described in the manuscript. This discrepancy suggest that neither the original papers nor this review provide much insight into how mitochondrial dynamics are mechanistically impacted.
  7. Figure 4 distinguishes between high and low nicotine levels but the text provides no background.
  8. The role of CS for mitochondria could be easier to describe mechanistically but the review does not mention how this occurs.

Minor points:

  1. The authors “depicted” mitochondrial dynamics in Figure 1 but did not “demonstrate” it.

Author Response

It is with excitement that we resubmit to you a revised version of manuscript cells-1708425, the manuscript entitled “Alterations of Mitochondrial Network by Cigarette smoking and e-Cigarette Vaping” for the Cells journal. Thank you for giving me the opportunity to revise and resubmit this manuscript. We appreciate the time and effort that the editor and the reviewers dedicated to providing suggestions for improvement. We have revised the manuscript, and as a result of the reviewers’ recommendations which we addressed, the manuscript has been dramatically improved. We highlight in yellow the changes that were made to the manuscript. Please see below for a point-by-point response to the reviewers’ comments and concerns.

Reviewer 2:

Kanithi et al. review the evidence that CS and ECV alter mitochondrial dynamics and ROS output. While the evidence in the case of CS is clear, less is known about ECV. Mitochondria are a main control unit of cellular redox and oxidative stress, making them particularly vulnerable to toxic agents found in either way of nicotine consumption and possibly nicotine itself. Mitochondrial dynamics are one possible readout of mitochondrial damage.

The fission and fusion of mitochondria is a clearly defined cell biological mechanism that has mechanistic triggers and inhibitors. Other than poorly described phenomenological observations, the review currently does not provide much insight into how these are altered and how likely that these pathways are altered in users. The authors present a novel topic that is of interest to the public and scientists but their text barely scratches the surface of the cell biology that it aims to review. Generally speaking, the review should distinguish the molecular effects of nicotine versus the effects of additives. And it must describe the mechanisms behind it, including concentrations used for individual observations. This could put them into context into what happens in users. In its current form, the review is of little use to readers who would like to know more about the mechanistic background. 

Specific points:

  1. Regarding any effects of nicotine on mitochondria, it should be mentioned whether the compound was added at concentrations resembling what is found in consumers.

Response: The concentration of nicotine consumed varies according to the company and brand of tobacco used.  None of the studies mentioned whether or not the concentration studied resembles what’s found in consumers. Thank you.

  1. The cell biological mode of action of nicotine is not described. How does it interfere with mitochondrial pathways? How does it trigger Mfn1/2 degradation? It is mentioned that it acts on nAChRs, but this is unlikely causing effects in mitochondria. This is the core of the review, but the text provides little to no information.

Response: We greatly apricate the reviewer for this comment. The cell biological mode of action of nicotine is well described in several reviews. This review focuses on the recent findings in CS and ENDS invitro and in vivo. We discussed the aberrant mitochondrial network proteins triggered by CS in figure 2 (after removed 2nd figure). We also mentioned text line lines from 242 to 247.

  1. The toxicity of certain flavors in ECV is mentioned. Are these present in sufficient amounts in users? The review proposes strict regulation but does not give much insight into what was tested and why mitochondria are impacted.

Response: Studies were primarily focused on a preliminary understanding of the cytotoxicity of the flavors and, most, unfortunately, did not discuss if these amounts are reflective of user consumption. Abouassali et al. did say that mice were exposed to e-vapor puff within the limits of consumer exposure, so that study was expanded on with more detail.

  1. The presence of selenium and arsenic is mentioned in vaping solvents. Are these always present, or are they occasional contamination?

Response: An additional resource was added to address the variability in metals/metalloids in vaping solvents. This variability and lack and lack of consistency is one of the reasons why regulation is needed, with regards.

  1. What detrimental changes in offspring are observed with ENDS? Why? Are they restricted to ENDS containing copper microparticles? How often are these used by manufacturers?

Response: The study discussing copper microparticles was removed as it wasn’t focused on maternal health. Two additional resources were added instead. Currently, there is not sufficient data to explain the impact of ENDS on human offspring primarily due to the variability in consumption. A few murine models and the susceptibility of embryonic stem cells were discussed.

  1. Some tests result in mitochondrial hyperfusion, others in mitochondrial fragmentation. These are fundamentally different behaviors, suggesting different experimental inputs have been used but they are not described in the manuscript. This discrepancy suggest that neither the original papers nor this review provide much insight into how mitochondrial dynamics are mechanistically impacted.

Response: Thank you for raising the excellent comment. This review discusses how the toxic changed to mitochondria vary based on the chemical composition.  In order to further explain the discrepancy, more clarification was added in the conclusion. This includes how exposure time and type can impact the differences in behavioral changes.

  1. Figure 4 distinguishes between high and low nicotine levels but the text provides no background.

Response: Thank you for raising the comment. Some EC refill fluids called do-it-yourself EC productions a large amount of nicotine, and we mentioned this in line number 541. now we included the text about high and low concertation of nicotine along with EC.

  1. The role of CS for mitochondria could be easier to describe mechanistically but the review does not mention how this occurs.

Response:  We discussed the occurrences by inhalation of SC and ENDS in the discussion. We emphasized the recent findings on the type of treatment in vitro.  

Minor points:

  1. The authors “depicted” mitochondrial dynamics in Figure 1 but did not “demonstrate”

Response: We demonstrated mitochondrial dynamics on pages 2 and 3, lines 84 and 108. We also mentioned Figure 1 in lines 110 and 113.

Thank you.

Round 2

Reviewer 2 Report

The authors have addressed almost all of my concerns. Only one spot requires clarification. 

Minor point: 

In the section where they mention that 10uM Nicotine cause the disappearance of Mfn1/2, the authors should specify the mechanism behind this observation. If not known they should state that this effect is based on an unknown mechanism. 

Author Response

Reviewer’s comment: In the section where they mention that 10uM Nicotine cause the disappearance of Mfn1/2, the authors should specify the mechanism behind this observation. If not known, they should state that this effect is based on an unknown mechanism.

Response: In that study, they did not find the exact mechanism involved in MFN1/2 disappearance induced by Nicotine. Moreover, in conclusion, they mentioned that “In future studies, it will be necessary to investigate the precise mechanism involved in nicotine-induced Mfn degradation.” We included the below text in line 237 as suggested by the reviewer and highlighted it with green color. Thank you.

 This effect is based on an unknown mechanism.
